# Chronic Occupational Exposure to Traffic Pollution Is Associated with Increased Carotid Intima-Media Thickness in Healthy Urban Traffic Control Police

**DOI:** 10.3390/ijerph20176701

**Published:** 2023-09-01

**Authors:** Abdulrazak O. Balogun, M. Margaret Weigel, Edmundo Estévez, Rodrigo X. Armijos

**Affiliations:** 1Department of Safety and Occupational Health Applied Sciences, Keene State College, Keene, NH 03431, USA; abdulrazak.balogun@keene.edu; 2Department of Environmental & Occupational Health, School of Public Health, Indiana University-Bloomington, 1025 E. 7th Street, Bloomington, IN 47403, USA; weigelm@iu.edu; 3Global Environmental Health Research Laboratory, Indiana University-Bloomington School of Public Health, Bloomington, IN 47405, USA; 4Center for Latin American & Caribbean Studies, Indiana University-Bloomington, Bloomington, IN 47405, USA; 5IU Center for Global Health Equity, Indiana University, 702 Rotary Circle, Indianapolis, IN 46202, USA; 6Centro de Biomedicina, Universidad Central del Ecuador, Quito 170129, Ecuador; leestevezm@gmail.com; 7Postgraduate Program in Public Health, Universidad Autónoma Regional de los Andes (UNIANDES), Ambato 180150, Ecuador

**Keywords:** traffic-related air pollution (TRAP) exposure, carotid intima-media thickness (CIMT), traffic police, Ecuador

## Abstract

Urban traffic officers in many low- and middle-income countries are exposed to high levels of traffic-related air pollutants (TRAP) while working vehicle control on heavily congested streets. The impact of chronic TRAP exposure on the cardiovascular health, including the carotid intima-media thickness (CIMT), of this outdoor occupational group remains unclear. This cross-sectional study compared the average mean and maximum CIMT measurements of two groups of relatively young, healthy traffic police (32 ± 7 years; 77% male) in Quito, Ecuador, who were without clinical evidence of serious cardiovascular or other disease. Previously published background data on PM_10_ (a TRAP surrogate) indicated that street levels of the pollutant were several orders of magnitude higher at the street intersections worked by traffic police compared to those working only in an office. Accordingly, officers permanently assigned to daily traffic control duties requiring them to stand 0–3 m from heavily trafficked street intersections were assigned to the high exposure group (n = 61). The control group (n = 54) consisted of officers from the same organization who were permanently assigned to office duties inside an administration building. Mean and maximum CIMT were measured with ultrasound. General linear models were used to compare the CIMT measurements of the high exposure and control groups, adjusting for covariates. The adjusted average mean and maximum CIMT measures of the high exposure group were increased by 11.5% and 10.3%, respectively, compared to the control group (*p* = 0.0001). These findings suggest that chronic occupational exposure to TRAP is associated with increased CIMT in traffic police. This is important since even small increases in arterial thickening over time may promote earlier progression to clinical disease and increased premature mortality risk.

## 1. Introduction

Traffic-related air pollution (TRAP), a major global public health and ecological concern, consists of a complex mixture of pollutant gases (e.g., CO, nitrogen oxides (NOx)), volatile organic (VOCs) and polyaromatic hydrocarbon compounds emitted by motor vehicles from tailpipes, secondary pollutants formed in the atmosphere including particulate matter (e.g., UFP, PM_2.5_, PM_10_), ozone (O_3_), evaporative emissions from vehicles, and non-tailpipe exhaust (road dust resuspension, brake pad and tire wear, road surface abrasion) [1,2]. TRAP is a significant contributor to poor air quality in many urban airsheds, where it accounts for an estimated 5–61% of ambient particulate matter [3], 45% of NOx, and 10% or less of VOCs [4].

Exposure to TRAP is associated with the triggering and/or worsening of hypertension, ischemic heart disease, myocardial infarction, and stroke, as well as increasing all-cause and circulatory system mortality risk [1,5,6]. Chronic exposure to TRAP also has been linked to atherosclerosis, a major underlying cause of cardiovascular disease (CVD) [7,8,9]. Atherosclerosis is a low-grade systemic inflammatory disease [10] characterized by endothelial injury and dysfunction, and the accumulation of lipids and fibrous material leading to the thickening of arterial walls over time [11,12,13].

Carotid intima-media thickness (CIMT) is an indicator of arterial injury (arteriopathy) that is often used to identify and track the progression of atherosclerosis [7,14]. A number of published studies have suggested that chronic exposure to TRAP, TRAP pollutant constituents [15,16,17,18,19,20,21,22], and/or close residential distance to traffic [22,23] is associated with greater CIMT in healthy adults, although others reported finding only a marginal or null association [24,25,26,27]. 

Traffic police are an outdoor occupational group that have high chronic exposure to TRAP. Prior studies investigating the health effects of TRAP exposure have focused on the respiratory effects [28,29,30,31,32,33,34], mutagenic and carcinogenic effects [35,36,37,38,39], cytotoxic effects [40,41,42,43,44], chromosomal damage [45], lipid metabolism [46,47], inflammatory effects [48,49], cardiovascular effects [50,51], nephrotoxicity [52], exposure to toxic metals [53,54,55], and physical and psychological hazards [56]. However, published studies of traffic police are lacking that examine the effects of chronic TRAP exposure on CVD biomarkers such as CIMT. This is an important limitation in the literature since even small increases in arterial thickening over time can potentially increase the risk for developing CVD and premature mortality. 

We conducted an analysis of secondary data to examine the association of chronic TRAP exposure on CIMT in working traffic police in a large urban center (Quito, Ecuador). We hypothesized that traffic officers permanently assigned to street-level traffic control duties (a proxy for high TRAP exposure) would show evidence of greater carotid intima-media thickness compared to officers assigned to indoor office work due to the pro-inflammatory effects of chronic exposure to high TRAP levels that promotes endothelial injury, leading to thicker arterial walls. 

## 2. Methods

### 2.1. Study Site

The study was conducted in the Quito Metropolitan District (QMD), the capital city of Ecuador. The densely populated urban center with a population of more than 2 million residents is located in a long (80 km), narrow (5 km), high-altitude valley (average elevation: 2850 m) between the eastern and western slopes of the Andes mountains [57,58]. The mountainous terrain restricts wind flow and promotes temperature inversions, trapping airshed pollutants [59]. Data collected by the QMD central air monitoring (CAM) network over the past two decades suggest that annual levels of PM_10_, PM_2.5_, and other criteria pollutants frequently exceed World Health Organization and Ecuadorian national standards [59,60]. 

The QMD’s high-altitude location reduces combustion efficiency and increases the amount of TRAP emitted by the 415,000 cars, trucks, and buses estimated as circulating daily in the city at the time of the study [61,62]. Many of these were fueled by diesel, an even “dirtier” fossil fuel than gasoline, i.e., 95% of trucks, 100% of buses, 8% of cars [63]. In addition, both the diesel (500 ppm) and gasoline motor fuels (2000 ppm) produced and sold in Ecuador have the highest sulfur content of any LAC-region country [63,64,65]. Furthermore, driving on hilly and mountainous roads in heavy urban traffic necessitates frequent braking and the fast wearing of brake pads and tires, resulting in even more TRAP release into the airshed [60,64]. Mobile sources are reported as responsible for more than one-half (52%) of the QMD’s greenhouse gas emissions and 46% of PM species [63,64,65,66]. 

### 2.2. Study Design and Participants

The present study analyzed secondary data from a larger study conducted from October 2009 to December 2010 investigating the cardiovascular health of QMD traffic police control officers. A convenience sample of traffic control officers from the QMD Traffic Control and Road Safety Operative Group (*Grupo Operativo de Control de Tránsito y Seguridad Vial del DMQ*) were recruited by the original study as prospective participants. These were officers who were permanently and exclusively assigned to either road traffic control duty or administrative office duty. To be considered for inclusion in the present study, prospective participants were required to be an active-duty officer who had continuously worked in the organization for at least four years. The other inclusion criteria included a negative medical history for chronic conditions (e.g., hypertension, other cardiovascular disease, diabetes, kidney disease, asthma, cancer) or infectious diseases (e.g., TB or other respiratory infections, HIV-AIDS, hepatitis) as determined by the internal medicine physician on the study team who interviewed them, reviewed their medical history charts from the police hospital, and performed a comprehensive physical examination on each. Prospective participants were informed that their participation was voluntary and they were provided with verbal and written explanations of the study objectives and its potential risks and benefits by the study team. 

A total of 300 participants were recruited for the main study. Ten were excluded from participation because their clinical examinations identified them as having an acute respiratory infection. The other 290 met the study criteria and were enrolled in the study. A subset of these (n = 115), or 35% of the enrolled participants, also underwent ultrasound evaluation of their carotid intima-media thickness (CIMT). Of these, 61 had permanent assignments to roadway traffic control (high exposure group) and 54 to the central administration office (control group). Both verbal and written informed consent were obtained from all participants prior to data collection. The study was carried out in accordance with the World Medical Association code of ethics (Declaration of Helsinki). The original study protocol was approved by the *Universidad Central del Ecuador* Biomedical Ethics Committee (protocol FA-05-0608; date: 6 October 2008). The findings reported in the current study were based on a secondary analysis of fully de-identified data used for dissertation research (AOB) performed at Indiana University (IU). The IU Institutional Review Board classifies studies based on de-identified secondary data as “research not subject to human subject regulations”.

### 2.3. Exposure

**High exposure group.** Road traffic police on the morning shift worked from 6:00 a.m. to 2:00 p.m., while those on the afternoon shift worked from 1:00 p.m. to 9:00 p.m. Regardless of what shift rotation they were randomly assigned to during a particular week, all road traffic police officers worked a period of three consecutive days at the same intersection until rotating to another for the next three days, then to another for three days, and so forth. The officers all stood in close proximity (0–3 m) to roadways while directing traffic but did not wear personal protective devices such as respirators. 

The major street intersections where they worked were characterized by a high mixed traffic volume that included commercial buses and trucks, taxis, and private cars, trucks, and motorcycles [67]. Traffic studies conducted at the same sites during the 4-year period (2007–2010) showed that traffic volumes ranged from 2110–11,352 vehicles/hour and from 2396 to 12,913 vehicles/hour during peak traffic hours in the same location [62,67]. Published background data from a 2008 ground-level air quality assessment campaign conducted by the QMD were used to estimate average PM_10_ levels (a TRAP surrogate) on heavily trafficked QMD roadways. These data were collected in and around many of the same streets in south, central, and north Quito where the traffic control police worked, including those transited by the ECOVIA public municipal transportation system and private and commercial vehicles (Figure 1). The published data indicated that the median PM_10_ levels measured at these traffic intersections was 1500 µg/m^3^ at the north and center ECOVIA sites, and 2100 µg/m^3^ at both the *La Marin* and *Necochea* sites, located in the center and south of Quito, respectively. However, after the installation of catalytic converters by the QMD on 42 municipal buses (April–December 2009), median PM_10_ concentrations decreased by more than half to 700 µg/m^3^ at the ECOVIA sites [60].

**Control group**. The traffic police assigned to office duties worked exclusively inside the central administration building during 8 h shifts that began at 8:00 a.m. and ended at 4:00 p.m. This administration building with natural ventilation was located on a one-way street in a residential neighborhood (see photo in Figure 2). We used published data to approximate the PM_10_ exposure of the control group who worked inside the central administration office. These background data were obtained from a prior study published on indoor PM_10_ values for administrative offices and classrooms in three QMD public elementary schools using a Harvard 5 LPM cascade impactor [63]. These ranged from 19.6 ± 13.6 µg/m^3^, to 20.4 ± 16 µg/m^3^, to 26.7 ± 15.9 µg/m^3^ [63]. The average indoor PM_10_ levels in these indoor spaces were several orders of magnitude lower than what the published data showed for the QMD street intersections worked by the traffic control officers [60,62]. 

### 2.4. Primary Outcome

**Carotid intima-media thickness (CIMT):** An experienced certified cardiologist with specialized training on ultrasound equipment and techniques conducted all CIMT ultrasound measurements at the AMCOR Center using the MicroMaxx 3.4.1 high-resolution M-mode digital ultrasound portable system linked to a 5–10 MHz multifrequency high-resolution linear transducer (Sonosite, Bothwell, WA, USA). The cardiologist was blind to the status of each participant as either traffic duty police personnel or control. 

Participants were placed in a supine position with the head extended and slightly rotated to the opposite side using a 45-degree head block [68]. The CIMT measurements were performed after the participant had rested quietly for 10–15 min. Semiautomatic measurements of mean and maximum CIMT were obtained using SonoCalc version 4.1 automated edge detection software (Sonosite, Bothwell, WA, USA). Thickness was assessed as both the mean and maximum of three predefined angles (anterior, lateral, and posterior) capturing the media-adventitia interface of the near and far arterial walls, 1 cm proximal to the bulb from both right and left carotid arteries. These mean and maximum values were recorded. The average mean CIMT for each participant was calculated using the following formula: (mean right CIMT + mean left CIMT)/2. The average maximum CIMT value was calculated as (maximum left CIMT + maximum right CIMT)/2.

### 2.5. Covariates

**Blood lipids.** Participants donated a fasting 8 mL venous blood sample for the blood lipid profile and other studies. The blood sample was analyzed in the Biomedical Research Center laboratory at the Universidad Central del Ecuador (UCE). Total cholesterol and triglycerides were analyzed using colorimetric enzymatic tests with lipid lightening factor (LCF) (CHOD-PAP and GPO-PAP, respectively). HDL cholesterol was measured using the precipitant technique. Measurements were conducted using an Eppendorf PCP 6121 Spectrophotometer (Eppendorf AG, Hamburg, Germany). LDL and VLDL cholesterol were calculated using Friedewald’s formula: LDL-C (mg/dL) = TC − HDL-C and TG/5, respectively [69]. 

**C-reactive protein.** A portion of the blood sample obtained from the participants was also used to measure high-sensitivity C-reactive protein (hs-CRP), a systemic inflammatory marker, at the UCE Biomedical Research Center laboratory. These analyses were performed using ultra-sensitivity immunoturbidimetric assay [70].

**Body mass index.** Participant body weight and height were measured by trained study technicians. Weight was obtained by weighing participants to the nearest kilogram without shoes or heavy clothing on a calibrated electronic balance at the police hospital. A stadiometer was used to measure participant standing heights to the nearest centimeter subsequent to removing any shoes, helmets, hats, or other headwear. The weight and height measurements were used to calculate body mass index (BMI), defined as weight (kg)/height (m^2^) [71]. The BMI measurements of <18.5, 18.5–24.9, 25–29.9, and ≥30 were classified, respectively, with underweight, normal weight, overweight, and obesity [71].

**Blood pressure measurement.** Blood pressure measurements were made with a calibrated manual sphygmomanometer following a published protocol [72]. Participants rested quietly for ten minutes in a seated position before three blood pressure readings were taken at one-minute intervals. The three systolic (SBP) and diastolic blood pressure (DBP) measurements were averaged and recorded. 

**Sociodemographic, smoking and family cardiovascular disease history characteristics.** Data on participant age, gender, family history of cardiovascular disease, any current smoking or alcohol use, and years of employment were collected from participants and cross-checked in the personnel medical records of the QMD traffic police. 

### 2.6. Data Analysis

The data were analyzed with IBM-SPSS (Released 2020. IBM SPSS Statistics for Windows, Version 28.0. Armonk, NY, USA: IBM Corp). All statistical tests were 2-tailed and the significance level was set at *p* < 0.05. The summary statistics are presented as number (%) or mean ± SD. We used Student’s independent *t*-tests or 2 × 2 contingency table analysis with X^2^, as appropriate, to compare between-group differences in means or proportions for the sociodemographic, lifestyle, clinical, and laboratory health indicators. We used general linear models (GLMs) to compare mean differences in the ultrasound-measured CIMT of the outdoor street traffic control officers (high traffic pollutant exposure group) with the indoor administrative office officers (control group). The model findings are presented as unadjusted and adjusted means with their respective 95% confidence intervals (CI). Covariates were considered for potential inclusion in the models based on their previously reported associations with CIMT in the literature. These included participant age (years), sex (male, female), current smoking (yes, no), family history of cardiovascular disease (yes, no), and measured systolic blood pressure, LDLc, and CRP. In this study, chronological age also served as a rough indicator of years of exposure, since the officers typically enter into the traffic control force after graduating from high school at age 18. 

## 3. Results

Table 1 compares the sociodemographic, anthropometric, clinical, and lifestyle attributes of the relatively young and healthy officers from the high TRAP exposure (street traffic control duty) and control groups (office work duty). As it indicates, the characteristics of the two groups were similar except for SBP, which was marginally higher in the administrative office group. 

Table 2 displays the findings from the unadjusted and adjusted GLMs comparing the mean and maximum CIMT measurements of the relatively young and healthy traffic police from the high exposure and control groups. As it shows, the high exposure group had significantly higher unadjusted and adjusted mean and maximum right and left CIMT measurements compared to those identified for the control group. In addition, the overall unadjusted mean CIMT (average of the mean right + mean left side) of the high exposure group was significantly thicker, i.e., 0.065 mm, than that of the control group. After adjusting for model covariates, the average thickness was 0.061 mm higher, i.e., was increased by 11.5% compared to that of the control group. The high exposure group also had a significantly higher unadjusted maximum CIMT (average of the maximum right + maximum left side) compared to the control group, i.e., a 0.075 mm increase. After adjustment, the difference was 0.071 mm, a 10.3% difference.

## 4. Discussion

This study compared differences in measured CIMT among relatively young, healthy police officers permanently assigned to street-level traffic control whose duties chronically exposed them to high levels of traffic pollution (high exposure group) with police officers from the same organization who were permanently assigned to indoor office work only (control group). Consistent with our a priori hypothesis, the high exposure group had significantly greater CIMT measurements compared to the control group. Specifically, we found that the adjusted mean and maximum CIMT measurements were respectively increased by 11.5% and 10.3% among the high exposure compared to the control group traffic police officers. This finding adds to the growing body of evidence documenting the proatherogenic effects of chronic exposure to ambient pollution in diverse populations. 

The lack of similar studies examining CIMT in traffic police precludes direct comparisons with our findings. However, they are consistent with prior reports investigating the association of chronic exposure to TRAP or individual TRAP pollutants such as PM_10_, PM_2.5_, black carbon, or NO_x_ [14,15,17,18,19], or residential distance to traffic or heavy traffic in adult [16,19,20,23] and child [61] populations with increased CIMT in non-occupational population settings worldwide. Prospective cohort studies are needed to confirm and expand upon our findings. These future studies should include traffic control officers with cardiovascular and cardiometabolic conditions, as the adverse effects of TRAP exposure on CIMT may be even greater. 

The findings from our study have high public health and clinical importance since even small increases in CIMT over time can accelerate faster clinical progression to cardiovascular disease. However, occupational exposure to TRAP is potentially modifiable through policy changes and interventions at the organizational level, e.g., mandating the use of personal protective devices, changes in work schedules. In addition, changes in urban planning policy, the strengthening and enforcement of clean air laws, use of cleaner technology, and other interventions can potentially reduce occupational exposures through the improvement of street-level TRAP levels and overall airshed quality [73,74]. 

The QMD has already adopted and/or is in the process of adopting environmental policies and interventions including motor vehicle mobility restrictions, installation of catalytic converters on existing buses, replacing the public transport fleet with clean mass transportation (metro system, electric buses), creation of low emission zones, growth and promotion of the bicycle lane network, technical rules for installing battery recharging infrastructure in public and private parking lots, creating Quito’s Climate Action Plan, natural areas conservation and reforestation projects, continuous improvement of Quito’s air quality monitoring network [65,74,75]. Because of the inflammatory nature of air pollutants from fossil fuel combustion sources, improving air quality will also have a positive effect on preventing other chronic conditions including COPD [76], asthma [77], chronic rhinosinusitis [78], chronic kidney disease [79], arrythmias [80], metabolic syndrome [81], type 2 diabetes [82], and cancer, among others [83]. 

## 5. Limitations

This study has some potential limitations which should be considered when interpreting its findings. One is that although the cross-sectional study design allows for inference about the association of chronic TRAP exposure with CIMT, it cannot prove causation. Another is that exposure misclassification could be a potential source of error. For technical reasons, our preliminary study had to rely upon previously published data on street-level and indoor office PM_10_ levels to assign group rather than individual-level exposures, the latter of which would have been preferable. Street-level monitoring is more realistic than neighborhood CAM, which tends to underestimate the exposures of officers standing 0–3 m from street traffic. However, even with roadway exposure data, there is likely some variation in the amount of traffic air pollutants between the different locations where police personnel were deployed. Personal air monitors would be a more accurate way of more precisely measuring individual exposure to pollutants, since this allows more precise determination of the dose–response relationship between pollutant exposure and CIMT. However, the deployment of personal air monitors for measuring particulate matter and toxic gases for each study participant was not practical at the time of the study (2009–2010) due to their high cost, bulkiness, and weight. Another potential limitation was the use of surrogate administrative office estimates for indoor PM_10_ exposures of police office workers. We had planned to use cascade monitoring equipment to collect measurements in the administration office building but the equipment experienced a technical failure, so we had to use a surrogate taken at the same time in another study. 

Another potential limitation is the possibility of unmeasured residual confounding from both traditional and non-traditional drivers of arterial damage and atherosclerosis such as that due to residential air pollution or noise exposures, sleep disturbances, physical inactivity, the microbiome, other stressors [26,56,84,85]. 

## 6. Conclusions

Police officers working street-level traffic control in the large urban centers of low- and middle-income countries are a high-risk occupational group due to their chronic exposure to high levels of TRAP pollution. Our study findings suggest that chronic occupational exposure to TRAP is associated with increased CIMT in even relatively young and healthy traffic police. This is important since even small increases in arterial thickening over time may promote their earlier progression to clinical disease and increase their risk for premature mortality. Our findings underscore the importance of enacting policy and interventions to mitigate the TRAP exposures of this occupational group. Prospective cohort studies should be conducted to confirm and expand upon our findings.

## Figures and Tables

**Figure 1 ijerph-20-06701-f001:**
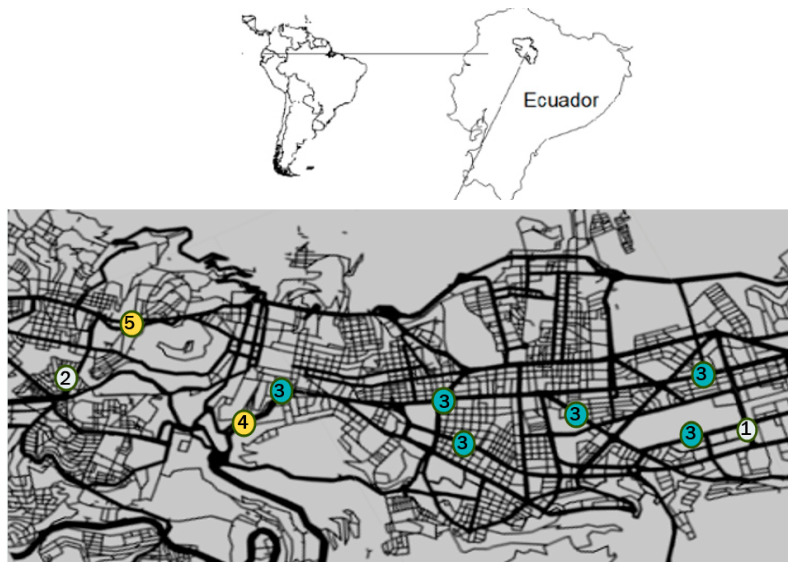
Map showing the QMD Traffic Control and Road Safety Operative Group working zone that is approximately 22 km long, extended at the north from United Nations Ave. (white circle 1) to Rodrigo de Chavez Ave. at the south (white circle 2). ECOVIA (green circles 3), La Marin, and Necochea (yellow circles 4 and 5, respectively). Map by R. X. Armijos (Adapted from Google, SNAZZY MAPS, Roadie, by anonymous).

**Figure 2 ijerph-20-06701-f002:**
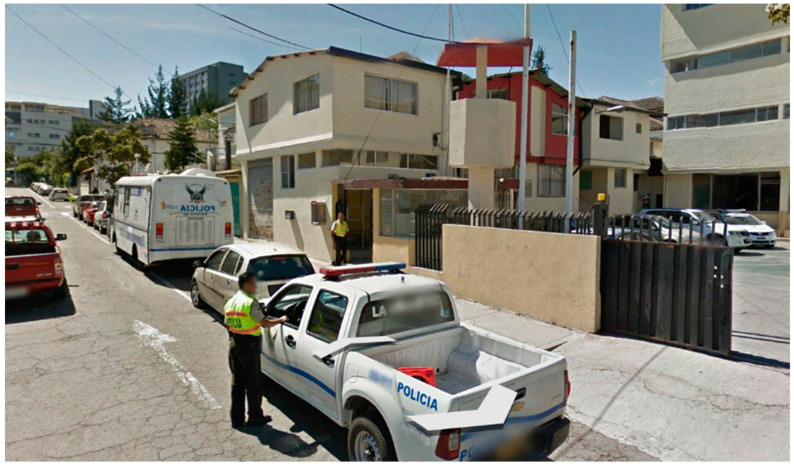
Central administration headquarters building of the QMD Traffic Control and Road Safety Operative Group located at the street intersection of Robles and Jose Tamayo (Source: Google Earth).

**Table 1 ijerph-20-06701-t001:** Participant sociodemographic and health characteristics stratified by exposure status.

	Total Sample (n = 115) Mean ± SD or No. (%)	High Exposure Group Road Traffic-Police Personnel (n= 61) Mean ± SD or No. (%)	Control Group Administrative-Duty Police Personnel (n= 54) Mean ± SD or No. (%)
Age (years)	31.6 ± 7.3	30.9 ± 8.4	32.4 ± 5.7
Sex/gender (male)	89 (77.4)	51 (83.6)	38 (70.4)
Body mass index	25.9 ± 3.5	25.9 ± 3.3	25.9 ± 3.7
Overweight	57 (49.6)	29 (47.6)	28 (51.9)
Obesity	13 (11.3)	8 (13.1)	5 (9.3)
Blood pressure			
Systolic BP (mmHg)	112 ± 12	110 ± 13	114 ± 10 *
Diastolic BP (mmHg)	73 ± 8	73 ± 9	74 ± 7
Blood lipid profile			
Total cholesterol (mg/dL)	210 ± 39	208 ± 39	212 ± 39
Triglycerides(mg/dL)	171 ± 103	178 ± 108	164 ± 98
HDLc (mg/dL)	49 ± 14	48 ± 11	51 ± 16
LDLc (mg/dL)	122 ± 29	124 ± 29	121 ± 29
VLDLc (mg/dL)	34 ± 21	36 ± 21	33 ± 20
C-reactive protein (mg/dL)	2.3 ± 3.2	1.9 ± 2.3	2.7 ± 4
Any current smoking	22 (19.1)	16 (26.2)	6 (11.1)
Any current alcohol consumption	35 (30.4)	20 (32.8)	15 (27.8)
Positive family history for cardiovascular disease	11 (9.6)	7 (11.5)	4 (7.4)

* *p* = 0.045.

**Table 2 ijerph-20-06701-t002:** Comparison of carotid intima-media thickness (CIMT) measurements by participant exposure status (n = 115).

	High Exposure Group Traffic-Duty Police Personnel (n = 61)		Control Group Administrative-Duty Police Personnel (n = 54)	
	Unadjusted Mean (95% CI)	Adjusted mean (95% CI) ^1^	Unadjusted Mean (95% CI)	Adjusted mean (95% CI) ^1^
Mean CIMT (mm)	0.537 (0.513, 0.561)	0.535 (0.513, 0.557)	0.472 (0.446, 0.497) *	0.474 (0.450, 0.497) *
Mean Right CIMT (mm)	0.546 (0.516, 0.575)	0.538 (0.510, 0.566)	0.459 (0.428, 0.490) *	0.467 (0.438, 0.497) **
Mean Left CIMT (mm)	0.528 (0.497, 0.559)	0.532 (0.501, 0.563)	0.484 (0.451, 0.517)	0.480 (0.447, 0.512) #
Maximum CIMT (mm)	0.695 (0.669, 0.721)	0.693 (0.668, 0.718)	0.620 (0.592, 0.647) *	0.622 (0.595, 0.649) *
Maximum Right CIMT (mm)	0.705 (0.674,0.737)	0.696 (0.666, 0.727)	0.628 (0.594, 0.661) **	0.638 (0.605, 0.671) ##
Maximum Left CIMT (mm)	0.685 (0.652, 0.719)	0.690 (0.656, 0.724)	0.612 (0.576, 0.647) ^	0.606 (0.570, 0.642) **

^1^ Analyses adjusted for participant age, sex, BMI, cLDL, positive family history for cardiovascular disease, systolic blood pressure, CRP, any current smoking, any current alcohol use; * *p* = 0.0001; ** *p* = 0.001; # *p* = 0.03; ## *p* = 0.02; ^ *p* = 0.004.

## Data Availability

The data presented in this study are available on request from the corresponding author.

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
