# Peer review of "Chronic Occupational Exposure to Traffic Pollution Is Associated with Increased Carotid Intima-Media Thickness in Healthy Urban Traffic Control Police"

_ijerph, 2023, doi:10.3390/ijerph20176701_

Round 1
Reviewer 1 Report
Line 194, the authors did not write the formula to calculate VLDL. Indeed it is included in the Friedewald formula the authors wrote; nevertheless, students will read this paper and people who are unaware of this topic.
Since the authors measured lipid profile, why did they not measure the apolipoproteins and use the Apo B/Apo A ratio, a better predictor of CVD risk (doi: 10.3390/metabo11100690).
Since the authors estimated the exposure concentration, it could be risky to make an association between exposure to TRAP and the CIMT. Moreover, they did not consider the amount of fast food, saturated fat, fruits, and vegetables both groups ingest. It is well known that those are significant variables that contribute to CVD.
Even though the limitations of the study, is there any statistical treatment to evaluate the degree of contribution of TRAP on the CIMT? The authors have the burden of ambient exposure of the police workers. Besides, the paper's title talks about an association between exposure to TRAP and CIMT, but no statistical test exists.
Author Response
Thank you for your comments, please see our reply in attachment.

Reviewer 2 Report
The paper is well-written.
Lines 52 - 56: The text needs to be rephrased for better clarity. It would also be good if the authors expanded the discussion of the prior findings of associations between adverse biological effects and exposures to traffic-related air pollutants.
Lines 103 - 111: Some subjects who may have been affected by TRAP could have been excluded. The authors decided to exclude subjects with a range of existing diseases. However, it is likely that TRAP exposure contributed to the development of those diseases. These subjects could have been affected by TRAP even more than those selected for the study. The authors should reflect the fact that "healthy" subjects were considered in the title, the abstract and elsewhere in the paper. I find this a major shortcoming of the study.
Line 142: Spell out ECOVIA.
The Conclusions should be written more clearly. The Conclusions lack information, as written.
Author Response

(The authors gave the same response as above.)

Round 2
Reviewer 1 Report
The authors improved the manuscript. They clarified the questions; the article can be published, congratulations!
Author Response
Thank you for your precious comments and acceptation.